# Emotional Transfer in Human–Horse Interaction: New Perspectives on Equine Assisted Interventions

**DOI:** 10.3390/ani9121030

**Published:** 2019-11-26

**Authors:** Chiara Scopa, Laura Contalbrigo, Alberto Greco, Antonio Lanatà, Enzo Pasquale Scilingo, Paolo Baragli

**Affiliations:** 1Italian National Reference Centre for Animal Assisted Interventions, Istituto Zooprofilattico Sperimentale delle Venezie, 35020 Legnaro (Padua), Italy; lcontalbrigo@izsvenezie.it; 2Department of Information Engineering, University of Pisa, 56122 Pisa, Italy; alberto.greco@unipi.it (A.G.); antonio.lanata@unipi.it (A.L.); e.scilingo@ing.unipi.it (E.P.S.); 3Feel-Ing s.r.l., 56122 Pisa, Italy; 4Department of Veterinary Sciences, University of Pisa, 56124 Pisa, Italy; paolo.baragli@unipi.it; 5Bioengineering and Robotic Research Center “E. Piaggio”, University of Pisa, 56122 Pisa, Italy

**Keywords:** human–horse interaction, animal assisted interventions, equine assisted interventions, horse, *Equus caballus*, emotional transfer, emotional intelligence, physiological signals, non-linear dynamic models

## Abstract

**Simple Summary:**

Equine assisted interventions (EAIs) represent an emerging field of animal assisted interventions (AAIs), employing horses in a wide variety of activities with humans. Based on the socio-emotional competences of this species, which evolved sophisticated communicative skills to interrelate with humans, we here hypothesized the occurrence of an interspecific emotional transfer during interventions. The emotional transfer hypothesis suggests a mutual coordination of emotional states of humans and horses, which are going through a coupling process during the interaction. Even though this mechanism is supported by few existing studies on human-horse emotional fine-tuning, it could play a key role in EAIs. We moreover propose to measure this coordination through monitoring physiological variables with a novel multidisciplinary method. In the future, having an insight on emotional states of animals involved in AAIs could be useful to ameliorate the wellbeing of both animal and human subjects and to better standardize operational strategies.

**Abstract:**

Equine assisted interventions (EAIs) include all therapeutic interventions aimed at improving human wellbeing through the involvement of horses. Due to the prominent emotional involvement traditionally characterizing their relation with humans, horses developed sophisticated communicative skills, which fostered their ability to respond to human emotional states. In this review, we hypothesize that the proximate causation of successful interventions could be human–animal mutual coordination, through which the subjects bodily and, most importantly, emotionally come into contact. We propose that detecting emotions of other individuals and developing the capacity to fine-tune one’s own emotional states accordingly (emotional transfer mechanism), could represent the key engine triggering the positive effects of EAIs. We provide a comprehensive analysis of horses’ socio-emotional competences according to recent literature and we propose a multidisciplinary approach to investigate this inter-specific match. By considering human and horse as a unique coupling system during the interaction, it would be possible to objectively measure the degree of coordination through the analysis of physiological variables of both human and animal. Merging the state of art on human–horse relationship with the application of novel methodologies, could help to improve standardized protocols for animal assisted interventions, with particular regard to the emotional states of subjects involved.

## 1. Introduction

Levinson [1] in his book *Pet-Oriented Science Psychotherapy* mentioned several examples of how animals could help in enhancing therapies with children. Levinson’s statement has been mentioned many times to implement the use of animals in therapeutic interventions and since the 1960s, this practice has become increasingly popular. Nowadays, animal assisted interventions (AAIs) are goal-oriented and structured interventions that intentionally include or incorporate animals in health, education, and human services (e.g., social work) for the purpose of therapeutic, educational, or recreational gains in humans [2]. According to the aim of the intervention, they are usually classified in animal assisted therapy (AAT), animal assisted education (AAE) and animal assisted activity (AAA), and they are structured and managed by a multidisciplinary team [3].

In this huge framework, equine assisted interventions (EAIs) are an emerging field, recently nominated as a very popular and novel practice [4]. EAIs is an umbrella term that includes a wide diversity of methodologies and approaches to improve human wellbeing through the involvement of horses (Equine assisted therapy—EAT; equine assisted education—EAE; equine assisted activity—EAA). EAIs can be adapted to the needs of the patient/beneficiary involved in a wide range of settings [5] and, for this reason, they are used in many institutions worldwide. In particular, equine assisted therapies (EATs) are often integrated in traditional therapeutic plans for individuals with mental and physical disabilities [6,7,8]. Even though their efficacy has not been completely proved yet, some authors claim social, emotional, physical, and educational benefits for several categories of patients experiencing therapy with horses [9,10,11,12,13,14]. For instance, EAIs seem to stimulate multiple domains of functioning handling emotional, cognitive, motor, and social disabilities with or without actual riding activity [15]. Kendall et al. [16] listed some anecdotal and descriptive hypotheses about how horses lead to positive psychological effects in patients during equine assisted therapy (EAT). Among these hypotheses, it has been suggested that EAT provides an emotional positive context that increases the prospect of beneficial changes in patients. For example, individuals with physical disability may experience a sense of “normality” while riding in contrast with the physical limitations they are used to face in everyday life [17]. In addition, therapeutic equine interventions are conceived to address self-esteem and personal confidence, communicative skills, and social trust, by literally making the horse a therapeutic tool [18]. By using unintentional signals (such as vocalizations or facial expressions, but also by seeking affiliative contact), humans and horses progressively sharpened the skills to communicate reciprocal affective states. The bonding process connecting humans with animals starts with physical contact. Information collected through the body are mainly used to anticipate the movements of the partner (both horse and human) [19]; however, body contact constitutes an emotional connecting channel between interactants as well [20,21], resulting in tangible behavioral and physiological variation. Therefore, human relationship with horses has been prompted by the emotional involvement consistently characterizing their interaction. The occurrence of repeated encounters in the long term is in fact useful for both motor coordination and socio-emotional engagement between the bonding subjects [19,22]. For this reason, most of the interventions imply physical interaction with animals.

The lack of a rigorous scientific approach in the study of these interventions results today as the main obstacle for the development of standardized methods in the field [23]. Here, we identify two main parallel branches in EAIs’ context both grounded on the same idiosyncratic process, i.e., the occurrence of coordination—*sensu lato*—between the interactants. A mutual interaction may in fact result in (I) a motor coordination dynamic or (II) in the coupling of physiological activities (brain/heart/hormonal) of both human and horse [24,25]. In the first case, the movement of the horse’s pelvis during horseback riding provides motor and sensory inputs to the human body producing normalized pelvic movement in the rider, closely resembling ambulation in individuals without disabilities [26,27]. Eventually, the rider’s motion becomes phase-matched with that of the horse, developing in a synchronized gait [28,29]. As for the second case mentioned, investigating horse-human interaction by simultaneously recording their physiological activities (such as heart rate or hormones levels) has been widely employed in the last decades, even though studies mostly focused on equitation disciplines or training [30,31]. Inter-subjects coordination is positively affected by the affiliative nature of the encounter since social interaction and the processing of affective information are suggested to facilitate the mechanism of embodiment (i.e., when body postures and facial expressions arising during social interaction play central roles in social information processing, [32]) [33]. Since direct human–horse relationship may significantly influence the emotional arousal of both individuals, consequently affecting their behaviors, physiological variables have been gradually incorporated in this field of study as easily accessible sources to evaluate the stress level or emotional condition of both humans and animals [34]. Moreover, it has been recently demonstrated that horse’s physiological activities overlap with the human’ ones, as long as the interaction occurs and that this convergence increasingly synchronizes when the interaction get “more intimate” [35,36]. Due to the prominent emotional involvement and the standardized methodologies characterizing animal assisted interventions, EAIs provide an attracting setting to test new approaches to study human–animal interaction. Scientific literature on this topic has been mostly focused on the human side, categorizing contexts and programs in which working with horses resulted effectively [37,38,39,40]. Nonetheless, those mechanisms implied in relating with animals that lead to beneficial effects on human life, have been a neglected topic in the recent scientific scene.

Based on these assumptions, the present review aims at getting at the root causation of the emotional human–horse coordination. Specifically, the main purposes are (a) to verify in which terms this process could be considered a cross-species emotional transfer and (b) whether physiological and emotional human–horse synchronization represents the key for successful EAIs. We carry out a comprehensive inquiry about how the emotional competences, co-evolved in these two species, fostered the progression of a potential inter-specific emotional sharing, based on the same neurobiological mechanisms.

Firstly, we listed few notional definitions of human–animal interaction, relationship and bond; later, we moved the focus on the concrete side of the process, from the merely physical up to the emotional aspects of interacting, from both human’s and animal’s perspectives.

## 2. From the Encounter to the Relationship: When Humans and Animals Interact

### 2.1. Human–Animal Bond: A Theoretical Framework

The human–animal bond has been defined as the mutually beneficial and dynamic relationship between humans and other animals, modulated by reciprocal behaviors that are essential to the health and wellbeing of both subjects involved [41]. This includes, but is not limited to, emotional, psychological, and physical interactions of humans, animals, and the environment.

In human psychology, the bonding process implies the establishment of a close, interactive relationship between the individuals involved [42]. Bonding typically goes in parallel with the concepts of attachment and affiliation, whose different degrees of intimacy determine the nature of the interaction. In particular: Affiliation is the simple act of being responsive with other human/animal subjects [43], but when affiliation requires social engagement and persistent reciprocal interaction with others, it develops into attachment [44]. According to this logic, bonding is the natural consequence of attachment and it is highly influenced by it in its essence, thus affecting the nature and quality of the bond itself. Bowlby formulated bonding and attachment theories in the sixties of the last century, integrating information that arose from his studies about child development and mother–child attachment. The ‘theory of attachment’ focused on the infant–caregiver relationship and its association to the healthy growth process. In his theoretical approach, Bowlby stated that childhood experiences lacking of a sense of safety and warmth, might influence the responsiveness of the adult in terms of stress and fear [42]. As for humans, in all mammal species the developmental phase is based on reliability in caregivers, whose nurturing behavior contributes to resilience in facing adversities later in life [45]. Even owners’ relationship with their pets can be described as parallel to the parental/child relationship [46]: The caregiver’s role in fact fulfills the people’ intrinsic desire to protect and, on the other hand, pets depend on caregivers for care and protection. This represents a strong component of the human–animal relationship.

Nevertheless, the classification of the multiple ways in which humans and animals come to interact is still under debate; moreover, it is difficult to force every interaction into an exclusive definition regardless of the animal species involved. In their review Hosey and Melfi [47] pointed out the slight differences between human–animal ‘bond’, ‘relationship’, and ‘interaction’ as previously defined by other scholars. In particular, the first to conceive a theoretical framework in this field was Hinde [22,48] who differentiated between the terms ‘interaction’, i.e., a sequence in which individuals show reciprocal behaviors to each other (which can be both positive or negative) and ‘relationship’ that implies the occurrence of a series of interactions over time. Eventually, Russow [49] listed some specific criteria that are necessary to outline a fully developed human–animal bond, such as reciprocity and persistence of the encounters. However, human psychological attributes, such as personality traits, empathy towards animals and people, human perception of pain in animals can also influence the interaction, consequently affecting the animal welfare and its cognitive performances [50,51]. Regarding human–animal attachment, in particular farm animals such as horses, three factors have been identified as having major impact on these animals’ ability and willingness to interact with humans: (I) The nature, quality, and frequency of contact with people, (II) the time period, and (III) the social environment in which it occurs [52].

In conclusion, the inter-specific bonding process clearly presents different characteristics if compared to the intra-specific one; however, they both appear to be based on the same essential mechanisms, i.e., reciprocity and emotional involvement.

### 2.2. How Horses Perceive Human World

To date the definition of human–animal interaction implied the occurrence of a dyadic encounter, in which the two subjects are recognizable and familiar to each other (as above stated). As properly discussed in Hausberger et al. [52], experience with humans in early months of life may have a great influence on horses, later in adult life. Such sensitive period of contact could allow a long-lasting reaction of the animal towards humans [53,54,55]. Some scholars proposed that early contact with foals could result in positive or negative association with humans, affecting the way in which horses relate to people in adult life and extending their experience to all human subjects [56,57]. It follows that reactions of horses towards people are the result of the interplay between the familiarities acquired with humans, the horses’ temperament, and the temperament and skills of humans [52]. One can speculate that, through the domestication process (about 2500–5000 years ago [58]), horses could have developed specific skills to relate with humans and that they consequently might have been selected for their ability to respond to human cues [59]. For example, horses are able to use human eye cues when deciding whether to follow a command or not. In particular, Sankey et al. [60] found that in response to a known vocal order horses obeyed with a similar rate to attentive familiar and unfamiliar persons (where the attentive state was evaluated based on the head orientation). Horses however monitored more the unfamiliar person’s attentional level, by turning their head and gazing at him. These results indicate a high individual representation level relying on the presence of sophisticated intra-specific cognitive skills. Since horses are able to form expectations on humans based on their behavior, body orientation/cues, attentional state, and past personal experience, they seem to have a “concept of person” as proposed in [60]. Horses have previously been shown to use human subtle cues such as gaze and body orientation, also when determining the focus of an attentive human [61]; still, when it came to use the same cues in an object choice task, horses employed the more basic mechanism of stimulus or local enhancement to get reward [62]. Thus, horses, similar to domestic goats [63] and contrary to dogs [64], are more likely to use the emitted human signal as a simple environmental stimulus rather than truly understanding the communicative nature of the cues provided. In their work, Lovrovich et al. [65] suggested that horses are indeed able to use humans as proximal local enhancement cue to find food. In this study, the tested subjects seemed to change their behavioral strategy to reach reward based on the experience acquired during the trials (by remembering the location of food hidden by the experimenter after a delay by using the position of the person, close to the target, as a valuable information). The ability to use human pointing indicating the location of hidden objects requires cognitive skills that go beyond the generalization of everyday interaction with humans. Pointing, along with any other way to get the attention of another to pursue a goal or reach something, is a form of referential communication whose success only depends on the audience’s attentiveness [66]. It has been proposed that referential communication have evolved in those species characterized by complex social systems, in which the use of visual signals play a pivotal role during interactions (for example, in fission–fusion societies, such as the horses’ one) [67,68]. Malavasi and Huber [69] investigated the possibility that horses would be able to apply referential communication at an interspecific level as well as at the intraspecific one. Authors here suggested that horses could adjust signaling according to the attentional state/presence of a human subject and that, in turn, these animals would be able to manipulate the human attention in order to achieve a goal. This presumed ability implies an overall understanding of the human’s different attentional states since horses seemed to behave differently depending on the degree of the experimenter’s attention.

It has been suggested for non-human primates that referential gestures arose from a ritualization of actions during inter-individual interactions [70]. Even though it is hard to assess whether these capacities are innate or rather they have been acquired through socialization with humans, the predisposition for intra-specific referential communication in horses can provide an advantage in relying on human visual communicative gestures when it is convenient. Likely, horses acquired these skills by individually interacting with people during repeated encounters.

In its original definition Hinde’s description of human–animal interaction was restricted to dyadic event thus implying that the two interacting members of the dyad should be recognizable and familiar to each other. In this regard, it is worth noting that horses can individually recognize not only conspecifics [71], but also familiar humans [72,73]. As happened for referential communication, individual recognition has been probably extended from only conspecifics to human subjects as well. In Proops et al. [71] subjects watched a herd member being led past them before the individual went out of view, and a call from that or a different individual was played from a loudspeaker positioned close to the point of disappearance. The tested subjects looked in the speaker’s direction significantly faster and for a longer time when the visual and auditory stimuli were not coherent (visual stimulus from an associate and auditory stimulus from a different one) compared to when they were coherent (both visual and auditory stimuli from the same associate). This is a clear indication that the incongruent combination violated their expectations. Concerning inter-species recognition, in [73] horses were presented with two people and then their voices from the speaker, which was at the center of the two people. Authors found that, when presented with familiar people, horses were capable to couple the voice with the person. In both these studies, the violation expectancy method was implemented to demonstrate that horses utilize the cross-modal recognition mechanism, i.e., the ability to integrate information perceived into some form of higher-order representation of a subject [74]. Therefore, horses are able to remember an individual (person or horse) thanks to their long-term memory and to recognize it later by matching different stimuli [69]. Discrimination of kin from non-kin, and of individuals within both categories, is probably the core ability underpinning the evolution of social behavior [75,76]; so then individual recognition is the most accurate way to categorize conspecifics. Regarding inter-specific recognition, it is reasonable to assume that horses have developed the capacity to recognize those humans with whom they interact due to the long-time co-evolution of these two species.

## 3. The Equine Social and Emotional Intelligence

Before analyzing the outcomes of the interaction with horses on human wellbeing, along with the variety of conceivable mechanisms through which it possibly works, it is crucial to carefully review the scientific literature about human–horse communication as the basis for their relationship. As previously mentioned, the bonding process and the development of a positive valenced long-term relationship needs a fertile field to emerge and progress, that is why not every animal can successfully interrelate with humans. Here, we listed those requirements horses should have and which are those that they effectively present, in order to assess their capacity to build relationships with human subjects. Following this path, it would be possible to determine the building blocks constituting the perceptive and communicative strategies underlying human–horse relationship (HHR) and in which measure horses and humans’ worlds overlap while interacting.

### 3.1. The Emotional Side of Human–Horse Relationship (HHR)

Domestic animals such as dogs and horses have shared many years of co-evolution with humans (about 5500 for horses), thus promoting the possibility to establish and grow inter-specific relationships. As pointed above, the main features of an effective human–animal relationship seem to be the exchange of reciprocal behaviors between the subjects involved and the occurrence of repeated encounters [22]. To comply both prerequisites, humans and animals evolved the capacity to communicate through a shared interface, which works as a cross-species common platform. The language used is mainly based on non-verbal signals, relying on (I) physical and (II) emotional connection. Physical contact and emotional reactivity represent the emotional channels connecting the subjects (see [77,78] for humans and [20,21] for animals). In their recent review, Payne et al. [21] gave an extensive summary on human–animal physical contact (I), underlined its pivotal role in bond formation according to the ‘human attachment theory’ [42,79,80]. In particular, petting and scratching have been found to actively reduce heart rate and fear indicators in horses [81] and dogs [82], even if provided by unfamiliar humans. Contrary to human–dog or human–cat interactions, humans and horses experience high level of body-to-body contact when engaged in an interaction. Even if horses are able to respond to a known vocal order coming from both familiar and unfamiliar humans [60], the body is the basis from which non-verbal human–horse communication grows, specifically in riding activities, in a sort of “kinesthetic empathy”(as defined by [83]).

On the other hand, emotional connection (II) fosters the bonding process between individuals due to the activation of a sophisticated mechanism of self-tuning its own emotions on others’ emotions. In humans, this skill called emotional intelligence (EI) [84], seems to influence inter-individual relationships since higher scores on emotional intelligence tests have been associated with various indicators of social adaptability [85] and with the development of emotional competencies [86]. Understanding emotional dynamics may help to anticipate one’s own and others’ emotional reactions and thereby to effectively manage emotions during a tense encounter [87]. With all due limitations of animal emotional competence, the possibility of the same strategy in human–animal relationship deserves further investigation. By using their emotional competence, horses could have evolved the capacity to foreseeing and accordingly reacting to the human’s emotional state. This ability to cope with emotions is likely to influence the emotional valence of the interaction as a whole. Mendl et al. [88] argued that animal discrete emotions could be represented in a two-dimensional space (already suggested in [89]); similarly, each emotional experience is valenced (a) as positive or negative, rewarding or punishing, pleasant or unpleasant, and it comes with a specific level of arousal (b) (from low corresponding to calm, to high corresponding to excited). As for humans, also in animals these subjective experiences come along with neural, behavioral, and physiological changes (facial expressions, activation of neural processes, heart rate variation) which can be objectively measured. In this perspective, depending on the perceived valence (positive/negative) of the encounter, the human–animal relationship could range from reassurance to fear, involving the activation of different cerebral processes that strengthen the positive or negative emotions induced [90,91]. Speaking about rewarding stimulation, seeking affiliative interactions is considered rewarding per se. Phillips [92] claimed the role of oxytocin in liking and the one of vasopressin in wanting and their unambiguous connection to pleasure in humans. In horses, social grooming has been found to reduce the groomer’s heart rate (i.e., a sign of relaxation and calming effect; [93]). So that, inter-specific relationships could be used as a mechanism to promote healthy neurobiological development through touching and proximity, evocating rewarding emotions (positive valenced) in both humans and animals. In this perspective, the quality of a relationship in a group acquires an adaptive value [94,95]. Furthermore, Hinde’s definition of relationship [35] suggests that the core of a successful interaction is the “positive” or “negative” valence of each interaction, which constitutes a step towards the next one; meaning that the nature of the first interaction determines expectations for subsequent encounters.

It has been investigated how horses are able to build a bond and keep positive long-term memory of humans when a positive reinforcement (i.e., reward, in this case food reward consisting of a few hand-given grain pellets) is associated to the interaction [57]. Not surprisingly, Baragli et al. [96] demonstrated that horses perceive and respond to humans based on their past interactions. In their report, Proops et al. [97] described how horses are able to form long lasting memories of specific human individuals only by the previous observation of these individuals’ subtle emotional expressions in pictures. Authors first presented horses with a photograph of a happy (positive valence) or angry (negative valence) face belonging to one of two human models; several hours later, the horses were presented with the same human subject previously shown in the photograph but assuming a neutral expression in this occasion. Results revealed a significant difference in the first gaze, with horses that had previously seen the angry face showing a left gaze (right cerebral hemisphere) bias when viewing the same live subject with the neutral face. In this case, horses remembered the identity of those individuals, which had been perceived as potentially harmful in the last encounter. This refined skill seems to allow horses to use the valence of human facial expression as a basis for future encounters with the same subject, building specific individual emotionally valenced memory to quickly detect intentions and emotional states. It could be argued that, as is the case of humans [98], in horses those emotions associated to pleasant events decrease in intensity less than the emotions associated to unpleasant events, thus reinforcing the memory of a positive interaction rather than a negative one. Animals share the same central and peripheral neural mechanisms involved in experiencing emotion in humans; for this reason, they will actively seek situations assumed to provide them with a pleasurable experience and avoid those that might be assumed to be negative. Therefore, it is likely that animals experience similar humans’ emotional states [99].

Finally, even though horses are considered prime examples of human companionship, little is known about horse–human attachment process, if we exclude the huge amount of research limited to equitation. The many aspects of human–animal interaction can be truly acknowledged only by conducting cross-analyses on social and emotional competences of both individuals involved.

### 3.2. Horses’ Perception and Communication of Emotions

In order to be effectively defined as ‘relationship’, human–horse interaction would need an additional feature, which is the occurrence of transfer of emotions underpinning stimuli, facial expressions, and vocal/non-vocal signals between the subjects. Emotional contagion occurs when the perception of emotion expression induces the same emotion in the receiver as in the producer of the signal and this mechanism is considered the basis of empathy [100,101]. Hence, the assessment of the ability in a given species to express and perceive emotion expressions starts with analyzing its emotional repertoire. Natural selection has fostered those behavioral strategies that promoted affiliative interactions and social stability, with emotional transmission enhancing higher coordination among group members and stronger inter-individual bonds [102]. Under natural conditions horses live in stable social groups [103] in which the within-members transmission of positive valenced emotions could contribute to group synchronization [104] and the rapid transfer of negative ones such as fear may, on the other hand, work as survival strategies for a prey species such as the horse [105]. As above explained, emotions are characterized by two dimensions: Valence (positive or negative) and arousal [88]. A signal performed by an individual could induce in the receiver both the same arousal level (i.e., contagion of emotional arousal) and the valence (i.e., contagion of emotional valence). Briefer et al. [106] evaluated if horses are able to decode the emotional valence and arousal of whinnies performed by familiar or unfamiliar conspecifics and whether any form of emotional contagion occurs. Results showed no clear evidence of contagion of emotional valence; nonetheless, authors demonstrated that horses reacted differently to separation and reunion whinnies when they are produced by familiar conspecifics, but no differences were found when unfamiliar individuals performed them, thus advocating the occurrence of emotional arousal transfer. Therefore, this study suggests that horses are able to convey emotional states using vocalizations and to perceive variation in vocal parameters accounting for emotional valence.

In addition to vocal signals, horses display a wide range of facial expressions [107]. In Wathan et al. [108] horses were presented with photographs representing facial expressions of their conspecifics captured in different contexts and their reactions were recorded. Results showed that perception of positive expressions elicited more approaching behaviors (positive valence) and decrease of heart rate (low arousal level) in tested subjects; on the other hand, negative expressions triggered avoidance behaviors (negative valence) and increase of heart rate (high arousal level). The same study was replicated to investigate whether horses enact the same mechanisms when viewing photographs of human positive (happy) or negative (angry) facial expressions [109]. As already discussed (see Section 2.2), results of this last study suggested that horses may have adapted an ancestral capacity to perceive and appropriately respond to emotional expressions of conspecifics and throughout their coevolution with humans, they may have extended this ability to communicate with morphologically different individuals, i.e., humans. Finally, in Nakamura et al. [110], horses matched human auditory (voice) and visual (facial expressions) stimuli, respectively performed by a speaker and a screen. In this case, horses’ heart rate increased when looking at a negative facial expression after hearing a positive voice (incongruent condition), indicating that cross-modal perception of human emotions occurs in a generalized form towards unfamiliar people.

Nevertheless, suggesting the occurrence of the contagion of emotional valence and/or arousal could be an overstatement in these cases, since authors used bidimensional representation of facial expressions (i.e., photographs or screen) which is for sure different then observing a live face. Moreover, Smith et al.’s [109] work has been highly criticized for the analyses of data and methodological approach (for extensive comments see [111]). These recent studies, however, shed light on the innate and acquired characteristics that horses deploy when interacting with other individuals. Regardless if they are conspecifics or humans, horses would be perfectly able to develop, manage, and keep a relationship with humans, creating a bond based on reciprocal emotional fine-tuning. After all, it could be argued that interacting with conspecifics mimics interacting with individuals of a different species.

## 4. Horses in Equine Assisted Interventions (EAIs)

Horses are the most involved animals among AATs, in particular with hippotherapy or therapeutic horseback riding sessions. The riding experience is often used in rehabilitation medicine as complementary activity to improve motor skills in children affected by cerebral palsy [112] and persons with spinal cord injury [113] and multiple sclerosis. Therapeutic activities with horses are widely used also in individuals with autism spectrum disorder [37]. Several positive effects have been reported for patients in terms of social, communication/language, and stress/behavior, as well as a reduction in autism symptoms [114,115]. The benefits to humans of equine assisted therapy have been well-researched also in post-traumatic stress and anxiety disorders [116]. Moreover, many studies claimed on the impact of equine assisted interventions as a whole on patients living with chronic illness [10] or eating disorders [117]. It is worth noting that the potential benefits of equine assisted psychotherapy and counselling have been studied as well, even if potential positive results [118] emerged along with negative ones [119]. The most comprehensive work analyzing the application of equine assisted interventions has been recently published by Stern and Chus-Hensen [120], considering both adults and children treatments across different conditions.

EAIs, as for all AAIs, are based on the emotional connection and evolving relationship between the animal, the patient/beneficiary, and the professional who provides the intervention. The animal subject should be considered as an integrated complement, who helps in building the connection between the patient/beneficiary and the therapist or the care professional who is managing the intervention [121]. In therapeutic setting, this connection is functional to the onset of therapeutic alliance (TA). TA has been proposed as a pantheoretical factor that accounts for all kinds of therapy with positive outcomes, regardless of the approach and methodology [122]. Therapist personal attributes such as being honest, flexible, respectful, warm, confident, empathic, and trustworthy contribute to a quick and positive development of TA, as well as the use of some techniques such as exploration, reflection, noting past therapy success, facilitating the expression of affect, and attending to the patient’s experience [123,124]. Some researchers pointed out the role of the animal in therapeutic setting as a mean for shaping or growing the positive nature of interpersonal relationship [125]. The animal acts as a social lubricant (see [126]), a facilitator of social interactions with other human beings, that helps the establishment of the bond between the patient and the therapist making the initial resistance easier to overcome and giving a safer perception of the environment [127,128]. This interpretation fits with one of the leading hypotheses regarding the benefits of human–animal interactions, which is the “social support hypothesis” [129].

A large body of literature supports evidences that the animal contact reduces psychological stress, increases social behavior in humans, ameliorates relational skills, and finally promotes positive attachment and resilience ability [9,130]. Beetz et al. [131] suggested in an exhaustive review, that the common mechanism underlying the positive physiological and psychological outcomes of both pet ownership and AAIs, is the activation of the oxytocinergic system (OTS). OTS positively affects hormones (e.g., cortisol), neurotransmitters (e.g., epinephrine, norepinephrine, and dopamine) and the autonomic nervous system reducing blood pressure, heart rate and heart rate variability, fear, and anxiety. Some studies documented a role of OTS in social bonds [132,133,134,135] and in emotional contagion [136,137], even though this last point needs more investigations. Moreover, this mechanism seems to promote TA [138] and it probably marked the AAIs’ success and popularity among therapists [139].

Indeed, a number of specialists encouraged the involvement of animals in therapy, not only to build an effective TA with the patient but also to use the relation between the patient and the animal as a tool to unlock delicate issues, such as unconscious worries and fears. For example, therapists could elicit discussions by pairing the patient to an animal that has undergone the same problem (for example, a person who has been physically abused may relate to an animal with an abusive past, projecting his emotions onto the animal) [2]. Moreover, animals are often laden with many different subjective meanings, since people use them to embody emotions or feelings that are both hard to express and likely to be repressed [140].

The subjective meaning that animated or unanimated objects assume depends on the type of interaction with the subject; therefore, the same object changes its proprieties according to the subjective universe: “All the properties of objects are actually nothing more than perceptual cues that are imprinted on them by the subject with which they enter into a relationship” [141] (p. 67). Von Uexküll argued that, when throwing an object at a dog, the object itself switches from a neutral characterization to a meaning-carrier element (a ball becomes a ball for play) as soon as it enters into a relationship with the subject (the dog). Regarding animated objects, animals regularly encounter animals of other species, including humans, thus attributing to these latter new meaning shortly after the interaction. By the same process, animals as well can be significant objects for humans, powering up only through interaction. The nature of previous interactions between horses and humans lead both to attribute a general significance, positive or negative, to each other that were mere neutral objects before the encounters [142]. The same mechanism may occur during EAIs, where both human and horse acquire a symbolic connotation. In addition, some therapists argue that because horses operate as members of a herd, they have evolved an elevated sensitivity towards others, which makes them active catalysts when engaging in social behavior [143].

### 4.1. Exploring the Hypotheses beyond Positive Outcomes of EAIs: The Emotional Transfer Hypothesis

According to the “social brain” hypothesis [144] the nature and complexity of social relationships reflects the cognitive demands for sociality, which in turn drove cerebral evolution in social mammals. Ungulates represents an ideal group to test theories about cognitive evolution since they display a huge variety of social behavior (ranging from almost solitary individuals, such as *Tapirus* [145], to large stable social group like in *Equus* [146]). Indeed, what is more surprising is that the total brain size is not significantly associated with group size in ungulates, but rather with social complexity in terms of inter-individual interactions on a regular basis and different types of relationships across group members. Even though social complexity was not the only factor related to large size of ungulates’ brain, this relation suggests that as the complexity of social bonds progressively increased the cognitive demands required to maintain these relationships, increased as well [147]. Following this interpretation, it is reasonable to assume that ungulates are naturally shaped to interrelate and to develop dynamic social systems; this skill could have been extended to humans, as they progressively tended to include horses in their natural social groups. Hence, horses may actually own the cognitive plasticity and behavioral flexibility required to manage elaborate relationships and they are likely able to bring into play these competences even when interacting with individuals of a different species. As already pointed out, the modality in which the human–animal coordination can occur during AAI could be conceptualized following two main branches: A mechanical and an emotional one. To date the mechanical line has been gradually more studied and exploited, for example measuring coordinated behavioral patterns (such as locomotor activity) during animal-assisted interventions. The interspecific cooperative side of the interaction, that arose from mechanical synchrony, meets some of the prerequisites also characterizing referential communication, subject–caregiver attachment, and a rudimentary socio-emotional intelligence. Along with mechanical, emotional coordination between the interacting human and animal could be approached as well, by analyzing involuntary responses in nonverbal communication. Whether the EAIs lie more on intentional or unintentional signals is hard to tell, even though one can speculate that, given the type of the interactions, these interventions have an emotional basis and an unconscious nature both belonging, by definition, to unintentional forms of communication. Most of the research on horses and equitation in particular, has been focused on how to maximize horses’ performance using intentional signals, but little attention has been given to unintentional ones. As above mentioned, horses are able to read subtle human cues together with human facial expressions and human emotions. Emotional states however come along with neural, behavioral, and physiological changes, resulting in measurable indicators [88,148], and a flourishing literature is showing how neurochemicals or physiological parameters can be helpful to understand emotional state of both humans and animals (i.e., cortisol level, heart and brain activity, blood pressure). Today’s challenge consists of approaching human–horse duo as a complex interaction of dynamical systems (DS). Specifically, DS are not stationary, hence with continuous modifications of their internal status over the time. Modeling horses and humans as DSs opens a new definition of the way in which the relationship can bear and evolve allowing to better understand this sophisticated inter-relationship.

In order to reach the best-coordinated performance, it would be desirable that non-verbal communication could be shared between subjects; meaning that the duo starts to work as one. The experience of synchronized behaviors has been associated to efficiency of bonding in humans [149] and, not surprisingly, synchronized neural activities seem to facilitate humans in assuming the mental and bodily perspectives of others, predicting their forthcoming actions [150], ameliorating communication by gestures [151], and facial expressions [152]. Crews [35] showed that brain maps of horse and human became more synchronized with increased interactions; in particular, horse electroencephalograph signals (EEG) gradually matched with those of human when passing from a total absence of contact (standing close to the horse), to contact (petting the horse), to active interaction (grooming the horse and sitting on it). Moreover, the brain maps indicated more synchronization with familiar human than with the unfamiliar one, pointing out the importance of the quality of the relation. In a recent study, Lanatà et al. [153] suggested a reciprocal affection of emotional state through human body odors. Human chemosignals in fact produced under concrete emotional conditions may prompt emotional stimulation in other individuals as well, triggering specific physiological parameters (human–human, [154]; human–dog, [155]). Results revealed that human body odors induce responses in autonomic nervous system in horses suggesting the possibility of a cross-species transfer of emotions. Researchers monitored electrocardiogram and cardiac activity (such as heart rate variability, HRV) in horses that have been tested with human ‘happy’ and ‘fear’ body odors’ samples. It seemed that emotionally-charged human chemosignals affect the physiological status of horses accordingly, suggesting that horses are able to “read” human emotional states via the olfactory perception system.

So then, since a successful cooperation cannot be accomplished without social synchrony in humans [156,157], the possibility that mutual influence has the same key role in inter-specific encounters as well should be considered. In AAIs, the success of the intervention itself is closely dependent on affiliative, trust-based bond and on the emotional involvement characterizing the human–animal dyad [158,159]; for these reasons, AAIs offer unspoiled occasions to investigate human–animal coordination, particularly on the emotional level. Indeed, human–animal interactions mimic other significant relationships in humans’ lives, combined enrichment environments, and affiliative channels characterizing non-verbal communication. Regarding the current scientific scene, except for few preliminary data investigating physiological synchronized activities towards specific targets (i.e., post-traumatic stress syndrome, [160]; at-risk youth, [161]; intellectual disability, [162]), human–horse emotional transfer remains a path worthy of exploring not only for its immediate applications but also for the potential role in the reconsideration of animals involved in AAIs.

### 4.2. Measuring the Emotional Transfer: Nonlinear Dynamical Methods Applied to Physiological Human-Horse Signs

Several studies, most of them conducted by veterinaries and animal physiologists, analyzed the heart rate of horses to investigate how they perceive human psycho-physiological state [163] and how mood changes could be transmitted from humans to horses under different handling and riding conditions [30]. An indicative example of human–horse emotional transfer is provided by the possibility of characterizing the effect of human actions, such as posture and voice, on horse autonomic responses and hormone/pheromone secretion [60,96]. Another exemplary study showed that horses experiencing discomfort were more aggressive toward humans and could present an increased heart rate and motor activity [164].

Nevertheless, a rigorous holistic vision of the interspecies emotional communication is yet to be found through an interdisciplinary approach, which is necessary to explore deeper the biological and behavioral basis of human–horse emotional relationship. More recently, a group of veterinaries, ethologists, and bioengineers [36,165,166], showed a multidisciplinary line of study integrating ethological approach with the theory of the dynamical systems (DSs) to investigate this complex relationship. Specifically, DS is a system whose state changes over time, and it is governed by a mathematical function that shows the time dependence. An analytical solution of the mathematical law allows to predict about the system’s future behavior (dynamical systems theory). DS theory has been applied to a wide variety of fields such as mathematics, physics, biology, chemistry, engineering, economics, history, and medicine [167].

In their hypothesis, humans and horses could be considered as two complex systems going through a coupling process during the interaction. This coupling is described by analyzing the cardiovascular dynamics through the study of the HRV signal. In fact, human and equine HRV can be considered as random variables [168], whose relationship can be analyzed by means of mathematical approaches, such as the weakly coupled oscillators (WCO) [169] and the information theoretic learning (ITL) [170]. Results of such approach showed that the major contribution to estimate the dynamical evolution of the human–horse relationship lays down the nonlinear interaction between the two systems. Specifically, a deep analysis of linear and nonlinear characteristics obtained by high order moments of the amplitude probabilistic distributions of HRVs were adopted with the aim of: (i) Investigating the statistical and temporal structure of the data (e.g., cross-corentropy, cross information potential, and correntropy coefficient); (ii) quantifying the amount of the coupling process in terms of coherence and synchronization (i.e., magnitude squared coherence, mean phase coherence [171]); and (iii) evaluating the nonlinear similarity of the HRV responses (i.e., dynamic time warping). Interestingly, when the interaction between humans and horses was more intense, a significant decrease of the similarity among the HRV time series was observed. This suggested that when humans and the horses were exposed to stressful situations, they react differently according to their own natural physiological pathway, which are known to be different in humans and animals.

Another finding evidenced that the emotional relationship among humans and horses is the results of a series of responses to eliciting stimuli, either coming from horse to human or vice versa. This phenomenon involves different perceptive channels such as the visual, olfactory, auditory, and tactile ones, which are under control of the autonomic nervous system. The key to accessing the emotional interspecies exchange is considering the whole of these autonomic responses with a multimodal, multisensorial, and multidimensional analysis. Consequently, this objective should include collaboration between mathematicians, physicists, engineers, veterinarians, ethologists, and physiology experts.

## 5. Conclusions

This review aimed at providing the key to understand the proximate causation of human–horse relationship, exploring its multifaceted nature from the equine assisted interventions’ perspective, in order to ameliorate the well-being of both the human and animal subjects involved. Starting from the investigation of those mechanisms required for a human–horse encounter to become a ‘relationship’, we defined the socio-emotional world of horses by reviewing the most significant studies on the topic. Eventually, we hypothesized that detecting emotions of other individuals and developing the capacity to fine-tune its own emotional state accordingly with that of others, may have fostered the success of equine assisted interventions, bringing positive effects on both sides. From the body-to-body contact (the most immediate aspect of interaction) up to the emotional transfer (the sophisticated process of connecting individuals through emotions), horses and humans became able to coordinate physiological activities through bonding, which subsequently increased the similarity in the way both perceive and experience their common world [172].

However, it has been often suggested that horses can “sense” the human mental state of mind when involved in EAIs; this misconception could generate questionable beliefs about horses’ capacity to empathize with suffering people. Instead, the horse is not supposed to be the principal caregiver of a patient, rather it represents the catalyst of the healing process with due regard for the animal’s welfare and need [113]. We hypothesized here that the efficacy of equine assisted interventions may lie in the capacity of horses to emotionally (and not only physically) interplay with humans, to such an extent that they eventually act together as a unique system. Emotional transfer and connectedness along with mutual beneficial effects of touching and physical proximity, may represent the backbones sustaining the relationship.

The fact that animals could have beneficial influence on people has been recognized from centuries; today it is well known that the deliberate inclusion of animals in a treatment plan lead to an “healing” effect on patients. Scientific research has widely investigated the “human side” of interventions, examining the modalities and the variety of applicability of this specific approach, but what the animal actually feels and which is the unequivocal mechanism who make the intervention effective are still open questions. In their work, Beetz et al. [131] reviewed several original studies on human–animal interactions and proposed the activation of the oxytocin system as the main cause of the psychological and physiological positive effects on human participants. This model states a reduction of stress-related parameters, an increased trust toward others, consequent reduced aggression, and enhanced empathy. Recently, it has been demonstrated that the existence of an oxytocin-mediated positive loop modulated by gazing between humans and dogs [173], puts a spotlight on animals’ perspective. Authors hypothesized that that human–animal bond has been promoted by a socially rewarding effect coming from sharing a common non-verbal language, confirming the effect of oxytocin also on the animal side of the couple. Moreover, such oxytocin-mediated loop seems to require the sharing of individual recognition of the partner. Since horses and dogs partly share same features in this case (such as individual recognition of familiar humans), this study offers a promising line for future studies on equines. Yet, research on oxytocin levels in both humans and animals are still quite rare, but the existence evidence clearly points to a bilateral positive effect of interacting, looking at both human’s and animal’s perspective.

According to Brofenbrenner and Evans’s [174] “bio-ecological theory”, the interacting process fosters the acquisition of skills and abilities of individuals later implemented to convey the behavior in many different developmental domains. Throughout a lifetime in fact, individual’s development takes place in enriched environments, promoting learning and social stimulation; this process does not rely on a unique factor but rather is the result of progressively more complex reciprocal interaction of the individual with other organisms and with the environment. Thus, interacting on regular basis over a certain period is what stimulates growth and change. We could speculate then that both human and horse acted as developing organisms to each other and that their enduring forms of interaction (from domestication onwards) may have contributed to the progression of a bilateral competence: The horse towards the human and vice versa. Animal assisted interventions, in which enriched environments and affiliative prolonged interactions are crucial parts of the practice, probably represents the best setting to observe the byproducts of this convergent human–animal evolution.

In conclusion, the basic concept of “biophilia”, defined as the interest in animals and in seeking a connection with them [175] underwent a widening re-interpretation during the last decades, witnessing a flourishing increase of scientific studies on the topic, more often focalized on the animals’ side. The next phase should be to bring more attention on the delicate area of animal assisted interventions, moving the focus on how human–animal duo works before getting to practical outcomes on ameliorating human health. In this huge domain, EAIs occupy a relevant niche, considering the peculiar evolutive pathway that horses shared with humans and the constant and growing presence of these animals in human lives.

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
