# Peer review of "Emotional Transfer in Human–Horse Interaction: New Perspectives on Equine Assisted Interventions"

_animals, 2019, doi:10.3390/ani9121030_

Round 1
Reviewer 1 Report
The submission 'Emotional Transfer in Human-Horse Interaction: New Perspectives on Equine Assisted Interventions' hypothesizes that mutual coordination between interactants, from both motor and emotional perspectives, may be the proximate causation of successful interventions in Equine Assisted Interventions (EAIs) programs.
Thus the article addresses an important gap in the existing body of literature that clearly identifies the roles of emotions but mostly relates them to humans and either way does not specifically look at the intricacies of the emotional human-horse relationship.
But EAIs, as for all AAIs, are based on emotional connection and evolving relationship between the animal, the patient/beneficiary and the professional who provides the intervention. This study specifically seeks to discuss if a cross-species emotional transfer may be integral to the process, and explain the level of co-evolution and communication qualities relevant for this hypotheses.
In doing so, the authors deliver a great overview into the current state of research.
Thus, my only suggestion is a rewording in the heading and abstract to clarify their contribution, especially seeing that the 'new perspective' really truly emerges on page 11 just before the conclusion. Or maybe it could be introduced earlier instead of building carefully (and maybe too cautiously) up to the 'new argument'?
I would like to thank the authors for their skillful writing and am looking forward to seeing this contribution in print.
Author Response
Reviewer #1
The submission 'Emotional Transfer in Human-Horse Interaction: New Perspectives on Equine Assisted Interventions' hypothesizes that mutual coordination between interactants, from both motor and emotional perspectives, may be the proximate causation of successful interventions in Equine Assisted Interventions (EAIs) programs.
Thus the article addresses an important gap in the existing body of literature that clearly identifies the roles of emotions but mostly relates them to humans and either way does not specifically look at the intricacies of the emotional human-horse relationship.
But EAIs, as for all AAIs, are based on emotional connection and evolving relationship between the animal, the patient/beneficiary and the professional who provides the intervention. This study specifically seeks to discuss if a cross-species emotional transfer may be integral to the process, and explain the level of co-evolution and communication qualities relevant for this hypotheses.
In doing so, the authors deliver a great overview into the current state of research.
ANSWER: We thank the reviewer for appreciating our work.
Thus, my only suggestion is a rewording in the heading and abstract to clarify their contribution, especially seeing that the 'new perspective' really truly emerges on page 11 just before the conclusion. Or maybe it could be introduced earlier instead of building carefully (and maybe too cautiously) up to the 'new argument'? I would like to thank the authors for their skillful writing and am looking forward to seeing this contribution in print.
ANSWER: We thank the reviewer for his/her valuable suggestions. We listed the main purposes of the review in the Introduction, hopefully clearer then before (lines 113-116). Since similar comment has been also done by Reviewer#3 we decided to rearrange the Abstract and the Introduction, emphasizing on emotional transfer hypothesis and promising new methodologies to measure it.
Reviewer 2 Report
The topic of this review is interesting and relevant for the journal Animals. This review is well written and highlights important points to understand how to objectively measure the human-horse emotional transfer as an explanation for the success of Equine assisted interventions. I have some suggestions to improve the paper.
I suggest that it will be useful to add a new paragraph investigating how to measure/ to maintain the welfare perspective of the horses used in such Equine assisted intervention. Indeed, some horses might be more adapted for such intervention. For example, horses have different personalities and some of them might be disturbed by such interventions.
L109: Please change to “imply”.
L394 : Could you please find another term for « vehicle ». It is not appropriate for the horses.
L422 : Please change “a stone at a dog” to “a ball for a dog”. Indeed, “throwing a stone at a dog” is a threatful and not a playful act.
Author Response
Reviewer #2
The topic of this review is interesting and relevant for the journal Animals. This review is well written and highlights important points to understand how to objectively measure the human-horse emotional transfer as an explanation for the success of Equine assisted interventions. I have some suggestions to improve the paper.
ANSWER: We thank the reviewer for his/her valuable suggestions.
I suggest that it will be useful to add a new paragraph investigating how to measure/ to maintain the welfare perspective of the horses used in such Equine assisted intervention. Indeed, some horses might be more adapted for such intervention. For example, horses have different personalities and some of them might be disturbed by such interventions.
ANSWER: We thank the reviewer for introducing this topic. As you can see, we added a couple of references on horses’ welfare (as suggested by Reviewer#3 as well): Mutual interactions between cognition and welfare: the horse an animal model, by Hausberger et al., 2019 and Why should Human-Animal interactions be included in research of working equid’s welfare? by Luna & Tadich, 2019 (line 159). Despite these papers do not entirely address the issue about welfare + Equine Assisted Interventions, they merge “horses’ welfare” with “Horse-Human relationship”. A new paragraph on welfare assessment of horses involved in EAIs would go beside the main point of our work, i.e. the mechanism that make EAIs successful (emotional transfer hypothesis). Since among the reviewers suggestions, it appears clear that the main problem of our work is that the focus is not clear, we would like to better smooth other collateral issues instead of deepen into them.
L109: Please change to “imply”.
ANSWER: It has been changed.
L394 : Could you please find another term for « vehicle ». It is not appropriate for the horses.
ANSWER: It has been changed.
L422 : Please change “a stone at a dog” to “a ball for a dog”. Indeed, “throwing a stone at a dog” is a threatful and not a playful act.
ANSWER: The example of the stone has been borrowed from von Uexküll’s work. In fact, the meaning is that any object, even a stone, could shift its nature from neutral to another (in this case, playful) when it enters into a relationship with the individual. However, we changed the sentence as follows: “when throwing an object at a dog, the object itself switches from a neutral characterization to a meaning-carrier element (a ball becomes a ball for play) as soon as it enters into a relationship with the subject (the dog).”
Reviewer 3 Report
Emotional Transfer in Human-Horse Interaction: New Perspectives on Equine Assisted Interventions
The idea to write a review on EAIs is very good and it would be of great interest for a broad audience.
At the general level, from the abstract it is not clear what type of manuscript the readers can expect, i.e opinion or a systematic review, and more importantly it is not clear what is the focus of this paper (mechanism of emotional transfer between the human and horses or establishing the evidence of the benefit of EAIs for the humans as well as the animals, or providing new methodological insight into how to advance the field). My feeling is that the overall organisation of the manuscript can be the main issue. Sections 3.1 and 3.2 along with the conclusion are quite solid paragraphs and to my view, are the core of this manuscript, supporting the idea of a beneficial relationship based on synchronised motor and physiological activation in the EAIs. The Dynamic Systems proposed here by the authors to assess the emotional transfer are not well described and again to me this looks a very crucial aspect to support the authors’ idea. I would, therefore encourage the authors to rethink the structure and main focus of this manuscript.
Below I have made some suggestions mainly on how to rephrase and make clearer some sections. I think the authors missed some recent references that can be relevant and important to cite (e.g. Mutual interactions between cognition and welfare: The horse as an animal model published in Neuroscience and Biobehavioral Reviews in 2019 by Hausberger et al., Assessing Equine Emotional States published in Applied Animal Behaviour Science in 2018 by Hall at al.,)
Lines 25 probably better to say “emotional regulation”
Lines 27-29 does not follow from the previous sentence.
Lines 72-74 I would rephrase this sentence because does not sound right.
Lines 74-93 From my understanding this paragraph is a bit the core of the manuscript and it will be the base of the following section at the general level but especially the first paragraph is not well introduced by the previous sentences and more importantly It is not clear why motor and physiological coordination/synchronization could be linked with the effectiveness of EAIs. Furthermore there are few terminologies like “simulation” or “embodiment” that cannot be understood from people that are not familiar with this or related field and it is worth to give some clarifications.
Line 94-96 If this is the aim of this review I would argue that EAIs is missing from the equation. Probably the authors want to see whether the emotional-matching state between the horses and the humans showed by the motor and physiological synchronisation is the main mechanism of the EAIs success.
Line 97 systematic here is used improperly unless some guidelines such as PRISMA has been used to search how the emotional competences have co-evolved between Human and Horse. Remove “tried” instead “We carry out”.
Lines 99-103 This sentence is way too long and the message that it supposed to give is unclear. Probably breaking it in several sentences might help. There is a same issue in the following three sentences.
Lines 110-112 This sentence does not read well.
Lines 114 115 I do not think “skills horses” here is appropriate. I would rephrase and keep it very simple. What is missing currently in the literature is the mechanism of the H-H relationship that leads towards the beneficial effect on the humans.
Lines 115-117 again the “sophisticated mechanism” probably is not appropriate word to use. Actually emotional contagion is a rather simpler mechanism with a huge evolutionary importance for social species (see for example de Waal FBM, Preston SD. Mammalian empathy: behavioural manifestations and neural basis. Nat Rev Neurosci. (2017) 18:498–509. doi: 10.1038/nrn.2017.72; Preston SD, de Waal FBM. Empathy: Its ultimate and proximate bases. Behav Brain Sci. (2002) 25:1–20. doi: 10.1017/S0140525X020 00018).
Lines 117-120 This sentence requires some work
Line 124 it would be good to add some references at the end of this definition.
Line 133 it is not clear to what precise prospective the authors here are referring.
Lines 137 it would be good to add a ref here.
Line 159 The conclusions here look good but whether they follow from the above paragraph argument is debatable. The operational model and structural mechanism have not been, for example well introduced to be able to fully appreciate the argument proposed.
Lines 217-222 The description of the cross model recognition study can be better described. Currently is rather convoluted and very difficult to follow.
Lines 226-227 It is not clear what is the main message
Lines 251-256 These sentences require some attention; currently they are very hard to understand.
Lines 271-273 What are these limitations? Plus in the following sentence the authors imply intentionality to the communicative interaction which probably is not always the case.
The dimensional approach mentioned in lines 276 was actually proposed by Russell in 1980. Furthermore I do not understand the connection between the dimensional approaches to the emotional transfer previously discussed. Mendl in 2010 tried to merge the discrete approach with the dimensional approach to better grasp the “core affect” of the subject.
Line 325 probably “express” is more appropriate than the word “produce”.
Line 355-357 I am sure that the criticism on the Smith et al (105) is not only limited to the type of stimuli used (picture vs face).
Nakamura et al line 358 tested the cross modal recognition of human emotions and not directly the emotional.
Line 438 I would suggest to add a few references to support the argument here.
Paragraph starting from 503 is well described.
Line 517 What are the theories of Animal Behaviour?
Author Response
Reviewer #3
The idea to write a review on EAIs is very good and it would be of great interest for a broad audience.
ANSWER: We thank the reviewer for appreciating our work.
At the general level, from the abstract it is not clear what type of manuscript the readers can expect, i.e opinion or a systematic review, and more importantly it is not clear what is the focus of this paper (mechanism of emotional transfer between the human and horses or establishing the evidence of the benefit of EAIs for the humans as well as the animals, or providing new methodological insight into how to advance the field).
ANSWER: As suggested by the other reviewers as well, the abstract has been rearranged and the Introduction as well, in order to clarify the work’s purpose.
My feeling is that the overall organisation of the manuscript can be the main issue. Sections 3.1 and 3.2 along with the conclusion are quite solid paragraphs and to my view, are the core of this manuscript, supporting the idea of a beneficial relationship based on synchronised motor and physiological activation in the EAIs. The Dynamic Systems proposed here by the authors to assess the emotional transfer are not well described and again to me this looks a very crucial aspect to support the authors’ idea. I would, therefore encourage the authors to rethink the structure and main focus of this manuscript.
ANSWER: We thank the reviewer for his/her suggestion, we modify the 3.1 and 3.2 paragraph in order to better explain the concept behind the new proposed approach.
Below I have made some suggestions mainly on how to rephrase and make clearer some sections. I think the authors missed some recent references that can be relevant and important to cite (e.g. Mutual interactions between cognition and welfare: The horse as an animal model published in Neuroscience and Biobehavioral Reviews in 2019 by Hausberger et al., Assessing Equine Emotional States published in Applied Animal Behaviour Science in 2018 by Hall at al.).
ANSWER: We thank the reviewer for his/her valuable suggestions. The references have been added (line 322, Hall et al., 2018; line 159 Hausberger et al., 2019).
Lines 25 probably better to say “emotional regulation”
ANSWER: It has been changed.
Lines 27-29 does not follow from the previous sentence.
ANSWER: Thanks for noting that. As said before and as suggested by other reviewers, we rearranged and changed the abstract.
Lines 72-74 I would rephrase this sentence because does not sound right.
ANSWER: It has been changed in the following way: “Nevertheless, the lack of a rigorous scientific approach in the study of these interventions results today as the main obstacle for the development of standardized methods in this field”.
Lines 74-93 From my understanding this paragraph is a bit the core of the manuscript and it will be the base of the following section at the general level but especially the first paragraph is not well introduced by the previous sentences and more importantly. It is not clear why motor and physiological coordination/synchronization could be linked with the effectiveness of EAIs.
ANSWER: We thank the reviewer for this consideration. We understand that a link between the part of the paragraph about the role of horses in EAIs and the part about the presence of coordination between interacting subjects is missing. We rearranged the paragraph, by moving up the part about why EAIs actually constitute a good field in which investigate emotional coordination. We here hypothesize that, beside the fact that human and horse tend to coordinate their motor and physiological activities, what also happens is the reciprocal coordination of emotional states. The evidence that proofs this hypothesis has still to be tested. We later suggest a mathematical approach on physiological signals, which could help measuring emotional coordination.
Furthermore there are few terminologies like “simulation” or “embodiment” that cannot be understood from people that are not familiar with this or related field and it is worth to give some clarifications.
ANSWER: We removed “simulation” and we gave an extensive explanation of “embodiment” since it sounds as the more appropriate term in the sentence.
Line 94-96 If this is the aim of this review I would argue that EAIs is missing from the equation. Probably the authors want to see whether the emotional-matching state between the horses and the humans showed by the motor and physiological synchronisation is the main mechanism of the EAIs success.
ANSWER: The sentence has been changed as follows: “Specifically, the main purposes are (a) to verify in which terms this process could be considered a cross-species emotional transfer and (b) whether physiological and emotional human-horse synchronization represents the key for successful EAIs”.
Line 97 systematic here is used improperly unless some guidelines such as PRISMA has been used to search how the emotional competences have co-evolved between Human and Horse. Remove “tried” instead “We carry out”.
ANSWER: The term “systematic” has been replaced by “comprehensive”; “tried” has been removed.
Lines 99-103 This sentence is way too long and the message that it supposed to give is unclear. Probably breaking it in several sentences might help. There is a same issue in the following three sentences.
ANSWER: The sentences have been changed as follows: “Through unintentional signals (such as vocalizations or facial expressions but also in seeking affiliative contact) humans and horses progressively sharpened the skills to communicate reciprocal affective states. Additionally, their relationship has been prompted by the emotional involvement characterizing the interaction. This bonding process, connecting humans with animals, starts with the physical contact. In fact, information collected through the body could be used to both anticipate the movements of the partner (both horse and human) and as an emotional connecting channel between interactants, resulting in tangible behavioral and physiological variation. Consequently, the occurrence of repeated encounters in the long term is useful for both motor coordination and socio-emotional engagement between the bonding subjects. This is the reason why most of the interventions imply physical interaction with animals”.
Lines 110-112 This sentence does not read well.
ANSWER: The sentence has been changed as follows: “Due to the prominent emotional involvement and the standardized methodologies characterizing animal assisted interventions, EAIs provide an attracting setting in which test new approaches to study human-animal interaction”.
Lines 114 115 I do not think “skills horses” here is appropriate. I would rephrase and keep it very simple. What is missing currently in the literature is the mechanism of the H-H relationship that leads towards the beneficial effect on the humans.
ANSWER: Thanks for your suggestion, the sentence has been changed as follow: “Nonetheless, those mechanisms implied in relating with animals that lead beneficial effects on human life, have been a neglected topic in the recent scientific scene”.
Lines 115-117 again the “sophisticated mechanism” probably is not appropriate word to use. Actually emotional contagion is a rather simpler mechanism with a huge evolutionary importance for social species (see for example de Waal FBM, Preston SD. Mammalian empathy: behavioural manifestations and neural basis. Nat Rev Neurosci. (2017) 18:498–509. doi: 10.1038/nrn.2017.72; Preston SD, de Waal FBM. Empathy: Its ultimate and proximate bases. Behav Brain Sci. (2002) 25:1–20. doi: 10.1017/S0140525X020 00018).
ANSWER: In this case, “sophisticated mechanism” was referred to emotional transfer, which it is probably much more complex then emotional contagion, since it supposes a reciprocal affection of emotional states, which eventually leads to a sort of synchronization/balance. de Waal himself in On the possibility of animal empathy (in Feelings & Emotions: The Amsterdam Symposium, 2003) proposed the Russian doll model of the evolution of empathy. Assuming that the emotional transfer hypothesis is right, for sure transfer of emotional states between two interacting individuals of different species, affecting their physiology and behavior, could be considered closer to outer layers of the doll then the inner ones. According to this model in fact the outer layers of the doll increasingly required emotion regulation, self–other distinction and complex cognitive skills, which we consider more sophisticated then emotional contagion (at the core of the Russian doll immediately after perception-action mechanism).
Lines 117-120 This sentence requires some work
ANSWER: The sentence has been changed as follow: “Firstly, we listed few notional definitions of human-animal interaction, relationship and bond; later, we moved the focus on the concrete side of the process, from the merely physical up to the emotional aspects of interacting, from both human’s and animal’s perspectives.”
Line 124 it would be good to add some references at the end of this definition.
ANSWER: This definition of human-animal bond comes from the American Veterinary Medical Association (reference number 40, at the end of the phrase). If the reviewer suggests an additional reference, we will integrate it in the definition.
Line 133 it is not clear to what precise prospective the authors here are referring.
ANSWER: It has been changed.
Lines 137 it would be good to add a ref here.
ANSWER: The appropriate reference for this sentence could only be Bowlby, 1969 (number 43), already cited before (line130) but also later. We moved the second reference (number 43) from line 143 to line 141.
Line 159 The conclusions here look good but whether they follow from the above paragraph argument is debatable. The operational model and structural mechanism have not been, for example well introduced to be able to fully appreciate the argument proposed.
ANSWER: We thank the reviewer for this consideration. What sounds misleading was probably the term “operational models”. We replaced it with the more simple “characteristics” since what we meant to say was simply that clearly the human-human relationship is different, at a practical level, from the relationship between a human and an animal.
Lines 217-222 The description of the cross model recognition study can be better described. Currently is rather convoluted and very difficult to follow.
ANSWER: The description of this experimental procedure was changed as follows: “In Proops et al. subjects watched a herd member being led past them before the individual went out of view, and a call from that or a different individual was played from a loudspeaker positioned close to the point of disappearance. The tested subjects looked in the speaker’s direction significantly faster and for a longer time when the visual and auditory stimuli were not coherent (visual stimulus from an associate and auditory stimulus from a different one) compared to when they were coherent (both visual and auditory stimuli from the same associate). This is a clear indication that the incongruent combination violated their expectations”.
Lines 226-227 It is not clear what is the main message
ANSWER: The sentence has been changed as follows: “i.e. the ability to integrate information perceived into some form of higher-order representation of a subject”.
Lines 251-256 These sentences require some attention; currently they are very hard to understand.
ANSWER: The sentence has been changed as follows: “To comply both prerequisites, humans and animals evolved the capacity to communicate through a shared interface, which works as a cross-species common platform. The language used is mainly based on non-verbal signals, relying on (I) physical and (II) emotional connection. Physical contact and emotional reactivity represent the emotional channels connecting the subjects”.
Lines 271-273 What are these limitations? Plus in the following sentence the authors imply intentionality to the communicative interaction which probably is not always the case.
ANSWER: Except for few studies on recognition of valenced facial expressions or whinnies, speaking of discrete emotions/emotional intelligence/understanding others’ emotions could be an overstatement referring to horses. That is why in our work we never use terms as synchronization/empathy/understanding other’s emotions; we can only hypothesize a mechanism analogous to others already investigated in different species when human and horse bond and interact. In our opinion, being conservative on these arguments is very important. Regarding the intentionality, the sentence has been changed as follow: “By using their emotional competence, horses could have evolved the capacity to foreseeing and accordingly reacting to the human’s emotional state”.
The dimensional approach mentioned in lines 276 was actually proposed by Russell in 1980.
ANSWER: We thank the reviewer for this suggestion; we decided to use Russell, 1980 (line 281) instead of Stanley & Meyer, 2009. It is in fact a more accurate quote.
Furthermore I do not understand the connection between the dimensional approaches to the emotional transfer previously discussed. Mendl in 2010 tried to merge the discrete approach with the dimensional approach to better grasp the “core affect” of the subject.
ANSWER: We thank the reviewer for this consideration. We here applied the definition of Mendl et al., 2010 of animal emotions firstly to introduce the concept of valence, which can affect behavior and physiology of both subjects (the one who experiences the emotion and the one who perceives it). Later in the same paragraph, in particular we talk about the valence of the encounter that could determine the interaction as a whole. Moreover, we applied the same definition by Mendl et al., 2010 (line 341) to explain a couple of studies investigating the emotional contagion of vocal (line 343, Briefer et al., 2017) or visual signals (line 352, Wathan et al., 2016) in horses.
Line 325 probably “express” is more appropriate than the word “produce”.
ANSWER: It has been changed.
Line 355-357 I am sure that the criticism on the Smith et al (105) is not only limited to the type of stimuli used (picture vs face).
ANSWER: We thank the reviewer for having noticed that. The reference here was in fact wrong; the study by Smith and colleagues we are referring to is Functionally relevant responses to human facial expressions of emotion in the domestic horse (Equus caballus), 2016 and it has been included in the bibliography. Therefore, regarding the content of this paragraph, the use of pictures as stimuli, appear as the main limitation. If the reviewer has additional suggestions on this, we will integrate them.
Nakamura et al line 358 tested the cross modal recognition of human emotions and not directly the emotional.
ANSWER: We understand that the reviewer here is referring to the emotional contagion. Therefore, we moved Nakamura et al., 2018’s work close to the other ones investigating the recognition of human emotions by horses. Later we stated that speaking about emotional contagion in these cases, as it has been done for horses-horses emotions’ recognition, would be an overstatement.
Line 438 I would suggest to add a few references to support the argument here.
ANSWER: References have been added.
Paragraph starting from 503 is well described. Line 517 What are the theories of Animal Behaviour?
ANSWER: What we wanted to say here is that it is possible to integrate ethology with other different approach such as Dynamical Systems’ approach. The sentence has been changed as follows: “showed a multidisciplinary line of study integrating ethological approach with the theory of the Dynamical Systems (DSs) to investigate this complex relationship”.
Round 2
Reviewer 3 Report
Emotional Transfer in Human-Horse interaction: new perspectives on Equine Assisted Interventions
The overall manuscript improved quite substantially. I made few other minor suggestions.
Line 23 how does the new proposed methodology ameliorate the wellbeing of the subject involved in the duo?
Line 30 does not read well
Line 31-33 I would rephrase as follows “We propose that detecting emotions of other individuals and developing the capacity to fine-tune own emotional states accordingly (emotional transfer mechanism), could represent the key engine triggering the positive effects of EAIs”.
Line 35-40 Very good text.
Line 75-80 I do not understand this section. There are very convoluted sentences which affect the flow of the text.
Line 84 I do not think “Nevertheless” is necessary.
Line 125 I was suggesting to move the reference (41 not 40) from the second sentence to the first sentence of the paragraph where the definition of the human animal bond is mentioned.
Line 351 Smith et al., 2016 this paper has been highly criticised especially for what concerns the analyses of the data. I would highly suggest to read the paper by Tim Schmoll https://royalsocietypublishing.org/doi/10.1098/rsbl.2016.0201. Schmoll’s criticisms are quite valid.
Author Response
The overall manuscript improved quite substantially.
ANSWER: We thank the reviewers and Editor for all the suggestions that improved our manuscript.
I made few other minor suggestions. Line 23 how does the new proposed methodology ameliorate the wellbeing of the subject involved in the duo?
ANSWER: What we meant here is that nowadays the emotional state of horses involved in AAIs has not been taken into account. So then, having an insight on horses’ experience during AAIs could be useful not only to make the animal more comfortable during interventions but also in finding the best way to make him interact with humans. Consequently, there could be positive effects for humans involved as well.
Line 30 does not read well
ANSWER: It has been changed.
Line 31-33 I would rephrase as follows “We propose that detecting emotions of other individuals and developing the capacity to fine-tune own emotional states accordingly (emotional transfer mechanism), could represent the key engine triggering the positive effects of EAIs”.
ANSWER: It has been changed as suggested.
Line 35-40 Very good text.
ANSWER: We improved the Abstract as suggested by the reviewers and we are glad if it is clearer than before now.
Line 75-80 I do not understand this section. There are very convoluted sentences which affect the flow of the text.
ANSWER: We thank the reviewer for this suggestion. We changed the sentence as follows: “The bonding process connecting humans with animals starts with physical contact. Information collected through the body are mainly used to anticipate the movements of the partner (both horse and human) [19]; however body contact constitutes an emotional connecting channel between interactants as well [20, 21], resulting in tangible behavioral and physiological variation. Therefore, human relationship with horses has been prompted by the emotional involvement consistently characterizing their interaction”.
Line 84 I do not think “Nevertheless” is necessary.
ANSWER: It has been removed.
Line 125 I was suggesting to move the reference (41 not 40) from the second sentence to the first sentence of the paragraph where the definition of the human animal bond is mentioned.
ANSWER: The reference has been moved as suggested. With the rearrangement of bibliography, the reference numb. 40 in the first version of the manuscript is now numb. 41 (the one by American Veterinary Medical Association).
Line 351 Smith et al., 2016 this paper has been highly criticised especially for what concerns the analyses of the data. I would highly suggest to read the paper by Tim Schmoll https://royalsocietypublishing.org/doi/10.1098/rsbl.2016.0201. Schmoll’s criticisms are quite valid.
ANSWER: We thank the reviewer for letting us know about this commentary. We added Schmoll’s work in our manuscript.